# IS SELF-SUPERVISION ENOUGH FOR TRAINING SENTENCE EMBEDDINGS?

## ABSTRACT

In NLP, sentence embeddings are crucial for many tasks such as information retrieval, classification, clustering, or visualizing collections of texts. Currently, top-performing sentence embeddings are derived from pre-trained language models that undergo extensive supervised fine-tuning. This contrasts with computer vision, where self-supervised training has demonstrated remarkable success. Here we show that self-supervision alone can produce high-quality sentence embeddings, albeit slightly below those from state-of-the-art supervised models. We systematically compare several existing augmentation strategies for positive pair generation in contrastive learning and show that text crops strongly outperform popular dropout-based augmentation. Using text crops, well-performing embeddings can be obtained even when training from scratch without using pre-trained model weights, or when training a bare token embedding layer without any transformer architecture. Overall, we show that self-supervised learning allows rapid training of text embeddings of a given dataset.

## 1 INTRODUCTION

Representing texts as vectors is important in natural language processing for both supervised (spam detection, sentiment analysis, semantic matching) and unsupervised (clustering, visualization, retrieval) downstream tasks. Such representations (or text *embeddings*) can be obtained with a wide range of methods, from simple bag-of-words representations such as TF-IDF (Jones, 1972) to transformer-based large language models (LLMs) (Zhao et al., 2023). These language models are trained with a token-level loss, and subsequent fine-tuning with a text-level loss is needed to obtain useful text-level representations (Xu et al., 2023). We refer to models and representations fine-tuned for representing entire texts as *sentence transformers* and *sentence embeddings*, following Reimers & Gurevych (2019).

In recent benchmarks (Muennighoff et al., 2023), sentence transformers relying on extensive supervised fine-tuning on large curated datasets have typically performed best, whereas self-supervised sentence training results in worse models — in stark contrast to computer vision, where self-supervised learning (SSL) has been immensely successful in producing semantically meaningful image representations (Balestriero et al., 2023). Various SSL approaches have been suggested for sentence representation fine-tuning — such as SimCSE (Gao et al., 2021) or DeCLUTR (Giorgi et al., 2021) — but it is unclear how their performance compares between each other and to the state-of-the-art (SOTA) embedding models.

Here we ask: is self-supervision sufficient for training sentence embeddings? We show that:

- self-supervised fine-tuning on a minimal amount of data (as few as 10 000 short input texts) can lead to large improvements in sentence embedding quality, achieving performance only slightly below supervised SOTA models.

- using text crops as positive pairs for SSL performs substantially better than other augmentations, including dropout-based augmentation used by SimCSE, contrary to some claims in the literature (Gao et al., 2021).

- most of the improvement during SSL fine-tuning is due to the generic sentence adaptation, with domain adaptation playing only a minor role.

- SSL fine-tuning based on text crops can yield reasonable embeddings even when training a model from scratch (without using a pre-trained LLM) and even when training a pure token embedding layer without any transformer architecture.

- unlike in computer vision, embedding quality peaks in the output layer, making projection heads and guillotine regularization (Bordes et al., 2023) unnecessary.

Our findings challenge the common belief that high-quality sentence representations require extensive supervised training on curated datasets, and provide some insights into the key aspects of self-supervised training of sentence embeddings.

## 2 RELATED WORK

Transformer-based language models receive a sequence of text tokens as input and produce a separate latent representation for each of the tokens as output (Vaswani et al., 2017). The BERT model (Devlin et al., 2019) and its variants such as RoBERTa (Liu et al., 2019), SciBERT (Beltagy et al., 2019), or MPNet (Song et al., 2020) include an additional classification token `[CLS]` to serve as a global representation of full sentences in downstream tasks. However, only a small fraction of typical BERT training is dedicated to sentence-level tasks, such that `[CLS]` sentence representations do not usually perform well at encoding sentence-level semantics (Thakur et al., 2021; Muennighoff et al., 2023; Jiang et al., 2022). Likewise, averaging all output tokens to obtain a sentence-level representation does not perform well either (Muennighoff et al., 2023).

To improve sentence-level representations, more sophisticated pooling strategies (Wang & Kuo, 2020) and post-processing techniques (Li et al., 2020; Su et al., 2021) have been suggested. Alternatively, a token-level model can be fine-tuned with a sentence-level objective, typically using contrastive learning (Xu et al., 2023). Here, pairs of similar texts are used as *positive pairs*, which are pulled together in the embedding space. Approaches differ in how similar texts are defined.

**In supervised contrastive learning,** positive pairs are collected based on some explicit notion of similarity. Sentence-BERT (SBERT) (Reimers & Gurevych, 2019) uses a curated dataset of paired texts such as question-answer pairs from Stack Exchange; their most recent model (2021) was trained on over 1 billion of such pairs. Similarly, Sentence-T5 (ST5) (Ni et al., 2021) and Sentence-GPT (SGPT) (Muennighoff, 2022) apply contrastive fine-tuning to T5 (Raffel et al., 2020) and GPT (Radford et al., 2018) models. For academic texts, SPECTER (Cohan et al., 2020) and SciNCL (Ostendorff et al., 2022) use citing and cited paper abstracts to form positive pairs.

**In self-supervised contrastive learning,** positive pairs are generated automatically from un-paired texts, similar to self-supervised learning in computer vision that relies on data augmentations (Chen et al., 2020). SimCSE (Gao et al., 2021) uses two different dropout patterns to form a positive pair of embeddings. This approach has also been used by Liu et al. (2021) and Yan et al. (2021), who additionally investigate other data augmentation techniques, such as randomly masking a part of the input text or shuffling tokens. Outputs of two distinct networks can also be used to generate positive pairs (Kim et al., 2021; Carlsson et al., 2021). Further, one can use adjacent chunks of a text as positive pairs; this was applied to train RNN (Logeswaran & Lee, 2018), GPT (Neelakantan et al., 2022), and BERT models (Giorgi et al., 2021; Izacard et al., 2022). Recently, synthetic generation of positive pairs has been explored leveraging generative LLMs (Zhang et al., 2023). While some recent work studied what happens to the representation during self-supervised fine-tuning (Jung et al., 2024), it focused on narrow supervised evaluation tasks and did not consider unsupervised downstream applications.

**In past benchmarks of sentence transformers**, models trained in a supervised way have been shown to outperform the ones trained with self-supervision (Thakur et al., 2021; Muennighoff et al., 2023). For example, SBERT's latest `all-mpnet-base-v2` achieved top results among all models of BERT-base size. On some tasks, it is outperformed by much larger models and by commercial embedding models like `text-embedding-3-large` from OpenAI and `embed-english-v3` from Cohere. Benchmarks of sentence transformers use various performance metrics, such as classification accuracy, nearest-neighbor query performance, or approximating ground-truth similarity between sentence pairs (STS benchmarks) (Agirre et al., 2012; Conneau & Kiela, 2018). However, some of them are known to yield conflicting conclusions (Muennighoff et al., 2023).

## 3 SELF-SUPERVISED CONTRASTIVE FINE-TUNING

### 3.1 DATASETS AND EVALUATION

To study self-supervised learning for sentence embeddings, we investigated how far a pre-trained language model could be optimized to produce useful sentence representations for a given dataset. For that, we chose a base model and fine-tuned it on a specific dataset of interest. Then, we conducted supervised evaluation on that same dataset, predicting class labels that were not used during the fine-tuning.

As a base model we used MPNet (Song et al., 2020; `mpnet-base`), following SBERT (Reimers & Gurevych, 2019; model `all-mpnet-base-v2` on HuggingFace). This model has `bert-base` architecture with 110 M parameters and uses 768 embedding dimensions. We used mean pooling over all tokens to obtain a single 768-dimensional output vector for each input text.

We performed our fine-tuning experiments on six datasets: the arXiv, bioRxiv, medRxiv, Reddit, and StackExchange datasets from the P2P clustering tasks of the Massive Text Embedding Benchmark (MTEB) (Muennighoff et al., 2023), and the ICLR dataset (González-Márquez & Kobak, 2024). The datasets differed in the number of samples (18–733 thousand) and classes (26–610; Table S1). Four of them comprised scientific abstracts from different disciplines, and the other two consisted of internet posts. While MTEB datasets have been publicly available and have often been used for benchmarking, the ICLR dataset has been assembled only more recently, ensuring that it was not part of the training data of any established supervised models (such as SBERT, SPECTER, or SciNCL). This is relevant for fair model comparison, because evaluation data leakage can lead to inflated performance estimates.

As our main evaluation metric we used $k$-nearest-neighbor ($k$NN) classification accuracy ($k = 10$ with Euclidean distance; we obtained similar values using cosine distance, see Table S2). This is a measure of local coherence: it is high if each paper's nearest neighbors belong to the same class. This metric is particularly relevant for data exploration applications, e.g. for visualisation or clustering, as many unsupervised learning algorithms rely on the $k$NN graph of the data. In contrast, linear classification accuracy does not convey how suitable an embedding is for data exploration. As an example, an embedding with high linear accuracy but low $k$NN accuracy (e.g. a single informative dimension and many uninformative ones) would yield poor visualisation and poor clustering, and would not be useful for unsupervised data exploration. $k$NN accuracy is similar to NN-based retrieval metrics (Muennighoff et al., 2023), but only requires class labels instead of ground-truth neighbors. Additionally, to validate the results obtained using $k$NN accuracy, we also evaluate the models on some of the MTEB tasks such as clustering, retrieval, reranking, and STS.

Note that we purposefully used the entire dataset first for self-supervised training and later for supervised evaluation. As SSL does not have access to class labels, this does not present overfitting issues. For supervised evaluation, we used a 9:1 train/test split (using only labeled data).

### 3.2 AUGMENTATIONS AND LOSS FUNCTION

We leveraged a contrastive learning approach analogous to SimCLR (Chen et al., 2020) for self-supervised fine-tuning. We compared different augmentation strategies for positive pair generation, including text crops (Logeswaran & Lee, 2018; Giorgi et al., 2021; Neelakantan et al., 2022), dropout-based augmentation (Gao et al., 2021), and variations of those (see Section A.1).

The cropping augmentation was set up as follows: for each input text $i$ in a minibatch of size $b$, we cropped out all possible chunks of $t = 2$ consecutive sentences (discarding all sentences under 100 and over 250 characters long) and sampled two chunks, one as the anchor text $a_i$ and one as its positive partner $p_i$. For example, if the abstract of our paper were in the training set, then one positive pair could look like this:

| In NLP, sentence embeddings are crucial for many tasks such as information retrieval, classification, clustering, or visualizing collections of texts. Currently, top-performing sentence embeddings are derived from pre-trained language models that undergo extensive supervised fine-tuning. | $\leftrightarrow$ | This contrasts with computer vision, where self-supervised training has demonstrated remarkable success. Here we show that self-supervision alone can produce high-quality sentence embeddings, albeit slightly below those from state-of-the-art supervised models. |
| --- | --- | --- |

Table 1: **SSL fine-tuning.** Values give $k$NN accuracy of the mean pooling representation in percent. Columns 1–5: off-the-shelf models. Columns 6–7: MPNet fine-tuned on each dataset using cropping and dropout augmentations. Columns 8–9: Embedding layer and full BERT model trained from scratch using cropping augmentations. Reported values should be interpreted with an error of up to $\pm 1\%$, corresponding to the binomial standard deviation $100\sqrt{p(1-p)/n}$ for test set size $n \approx 2000$ and accuracy $p = 0.5$.

| | (1) | (2) | (3) | (4) | (5) | (6) | (7) | (8) | (9) |
|---|---|---|---|---|---|---|---|---|---|
| Model | MPNet | SimCSE | SPECTER | SciNCL | SBERT | MPNet | MPNet | Emb. | BERT |
| Pre-trained | yes | yes | yes | yes | yes | yes | yes | no | no |
| Augmentations | — | — | — | — | — | Crops | Dropout | Crops | Crops |
| ICLR | 37.4 | 45.7 | 56.8 | 57.0 | 63.3 | 58.9 | 46.8 | 57.3 | 57.1 |
| arXiv | 37.8 | 40.0 | 44.2 | 45.2 | 46.2 | 44.2 | 39.9 | 43.4 | 44.3 |
| bioRxiv | 58.6 | 59.0 | 64.8 | 66.4 | 65.2 | 61.8 | 60.7 | 60.7 | 60.6 |
| medRxiv | 43.5 | 47.2 | 52.6 | 52.8 | 56.8 | 52.4 | 47.8 | 49.1 | 44.9 |
| Reddit | 62.6 | 59.9 | 55.2 | 57.3 | 75.0 | 72.0 | 57.8 | 63.6 | 61.8 |
| StackExchange | 39.3 | 40.7 | 41.5 | 42.9 | 50.6 | 45.6 | 41.6 | 45.2 | 45.4 |
| **Average** | 49.8 | 51.9 | 55.7 | 56.7 | 61.9 | 58.5 | 52.1 | 55.9 | 55.6 |

For the dropout-based augmentations, we used the approach of SimCSE (Gao et al., 2021). We split each input text $i$ into groups of consecutive sentences in the same way as for the cropping augmentation, to have similar text lengths for both augmentations. Then, we sampled one single crop and passed it through the model twice, with two different random dropout patterns applied to it, yielding two different representations that we used as anchor $a_i$ and positive pair $p_i$.

As negative examples for text $i$ we always used the positive partners of all other anchors within the same minibatch $\mathcal{B}$. Unlike some other recent studies, we did not use any *hard negatives* (Cohan et al., 2020; Giorgi et al., 2021; Ostendorff et al., 2022).

During contrastive training, the cosine similarity between the representations of $a_i$ and $p_i$ is maximized, while minimizing the cosine similarities between $a_i$ and $p_j$ for $j \neq i$ within the same minibatch $\mathcal{B}$. This can be achieved using the InfoNCE loss function (Oord et al., 2018), also known as the normalized temperature-scaled cross-entropy loss (NT-Xent) (Chen et al., 2020). For one sample $i$, the loss is given by:

$$\ell_i = -\log \frac{\exp\big(\text{sim}(a_i, p_i)/\tau\big)}{\sum\limits_{j \in \mathcal{B}} \exp\big(\text{sim}(a_i, p_j)/\tau\big)} \,, \tag{1}$$

where $\text{sim}(a, p) = a^\top p/\big(\|a\| \cdot \|p\|\big)$ is the cosine similarity. We set the temperature to $\tau = 0.05$ and the batch size to $b = 64$, the largest possible batch size given our GPU memory resources. We trained the network using the Adam optimizer (Kingma & Ba, 2014) with learning rate $\eta = 2 \cdot 10^{-5}$, with linear warm-up and linear decay. See Section A.1 for details on hyperparameter choices.

### 3.3 CROPPING AUGMENTATION STRONGLY OUTPERFORMS DROPOUT AUGMENTATION

Out of the box, MPNet resulted in representations with low $k$NN accuracies (Table 1) and almost no semantic structure visible in 2D visualizations, obtained using $t$-SNE (van der Maaten & Hinton, 2008) (Figure 1). Whitening MPNet's representation increased the performance slightly when using the cosine metric for NN search (Table S2).

After fine-tuning MPNet for only one single epoch, the quality of the embeddings markedly improved (Table 1). Out of the two augmentation strategies, cropping worked much better than dropout: across datasets, dropout on average led to only 2 percentage points improvement from vanilla MPNet, while crops improved the performance by 8 percentage points. In some cases, such as for the ICLR dataset, the difference was even larger, with $k$NN accuracy increasing by over 20 percentage points using crops, versus only 9 points using dropout. The performance of the off-the-shelf SimCSE model (trained using dropout augmentations) was almost identical to our results using dropout augmentations (Table 1). Similarly, $t$-SNE visualizations suggested that the embed-

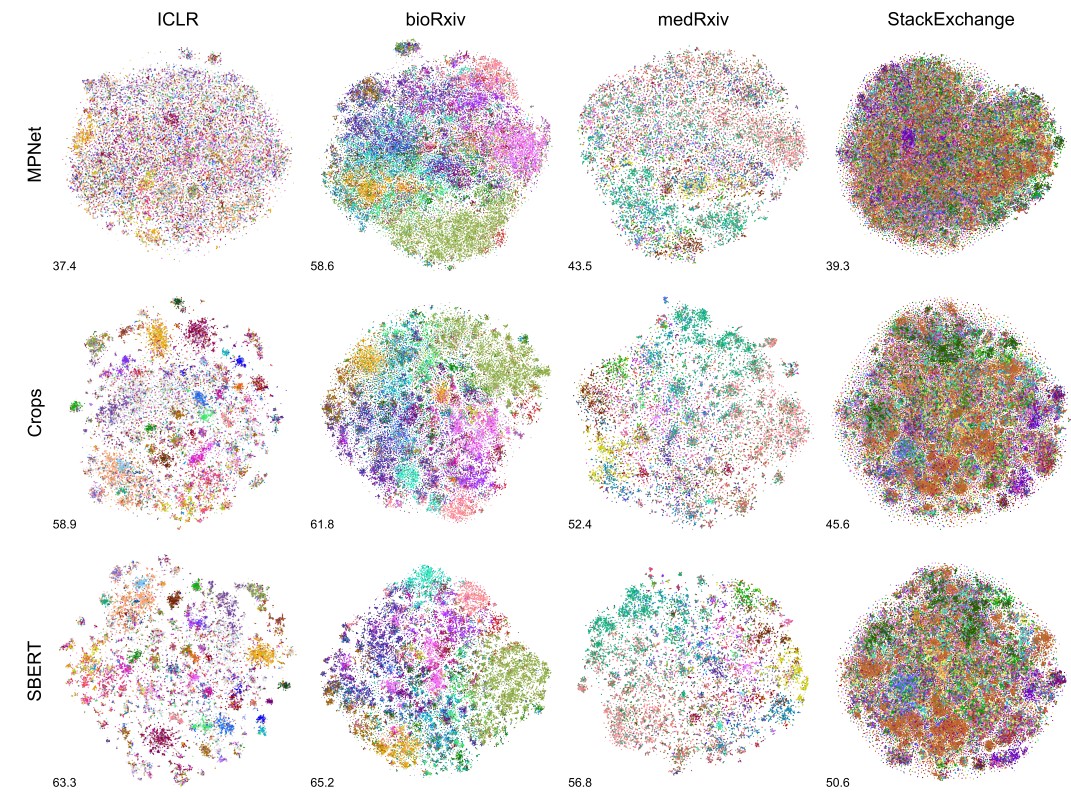

Figure 1: *t*-SNE visualizations of MPNet, SBERT, and cropping-based fine-tuned model embeddings of different datasets. Color corresponds to class labels. Numbers show *k*NN accuracy in 768D embedding space. We used openTSNE with default parameters (Poličar et al., 2024).

ding space after crop-based fine-tuning had clearly improved compared to vanilla MPNet (Figure 1). Fine-tuning beyond one epoch did not yield additional performance gains.

Notably, the embedding quality after crop-based fine-tuning was only 3 percentage points below SBERT (Table 1), which is the best performing sentence embedding model within models of its size (Muennighoff et al., 2023), and correspondingly, the *t*-SNE visualization was qualitatively similar to the representation obtained with SBERT (Figure 1).

Furthermore, on three scientific datasets (ICLR, arXiv, medRxiv), our cropping-based fine-tuning matched the performance of SciNCL and SPECTER, two off-the-shelf embedding models specifically designed and trained to represent scientific abstracts (Table 1, columns 3–4), using scientific citations as positive pairs. On the non-scientific datasets (Reddit and StackExchange), cropping-based fine-tuning unsurprisingly outperformed SciNCL/SPECTER.

To determine whether the token-level pre-training was necessary to achieve good sentence representations, we performed cropping-based contrastive training of the `bert-base` architecture from scratch, without using pre-trained MPNet weights (Table 1, column 9). Here, the performance did not saturate after one epoch, so we continued training for 10 epochs. On average across datasets, the resulting performance was only 3 percentage points below the one we obtained from a pretrained MPNet, and for four out of six datasets there was no noticeable performance difference at all.

Furthermore, to determine whether the `bert-base` architecture was necessary in the first place, we performed the same cropping-based contrastive training of a bare, randomly initialized, embedding layer. This is a direct token embedding model without any transformer architecture whatsoever. Training it for 10 epochs, we obtained embeddings that also were only 3 percentage points below full MPNet fine-tuning (Table 1, column 8). Again, for four out of six datasets there was almost

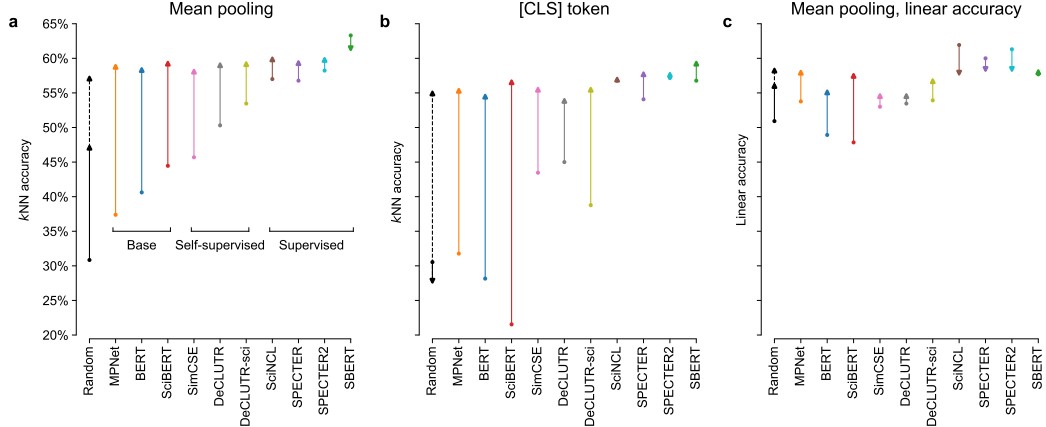

Figure 2: Evaluation of the representation quality before and after contrastive training. The arrows show increase in performance after one crops fine-tuning epoch. Dashed arrow indicates further increase after 10 epochs for the randomly initialized model. **(a)** $k$NN accuracy of the mean pooling representation. See also Table S3. **(b)** $k$NN accuracy of the `[CLS]` token representation. Note that here `[CLS]` representation was also used in the InfoNCE loss during the self-supervised training. **(c)** Linear accuracy of the mean pooling representation.

no difference at all. However, below we will show that generalization capability of the embedding layer model was much worse (Section 4.2).

Note in particular that the trained embedding layer always performed equally or better than MPNet with dropout fine-tuning, highlighting the large qualitative difference between the cropping and dropout augmentations. Dropout augmentations were not useful for sentence embeddings, while cropping augmentations could successfully train even a bare embedding layer.

### 3.4 CROPPING-BASED CONTRASTIVE TRAINING IMPROVES MOST OFF-THE-SHELF MODELS

To study whether the observed effects hold true for additional off-the-shelf models, we focused on the ICLR dataset, as it was the dataset with the largest performance gap between MPNet and SBERT (Table 1). We evaluated ten off-the-shelf pre-trained models (Table S3) and applied cropping-based fine-tuning to each of them. The publicly available models all used a `bert-base` architecture: base models (MPNet, BERT, and SciBERT), self-supervised sentence transformers (SimCSE, De-CLUTR, and its scientific version DeCLUTR-sci), and supervised sentence transformers (SciNCL, SPECTER and its newer version SPECTER2, and SBERT). In addition, we evaluated two commercial models: `embed-english-v3.0` in `clustering` mode by Cohere (1024-dimensional embeddings) and `text-embedding-3-large` by OpenAI (3072-dimensional embeddings).

All base models performed poorly and showed accuracy below 45%. Self-supervised sentence transformers exhibited accuracy in the 45.7–53.5% range. They were outperformed by the sentence transformers trained with citation supervision, showing 56.8–58.2% accuracy. SBERT showed best results (63.3%). The two proprietary models performed similar to SBERT: the model by Cohere with 62.9% $k$NN-accuracy, and the one by OpenAI with 62.3%.

After cropping-based fine-tuning, all models, except for SBERT, improved and reached very similar final performance with 58.1–59.8% $k$NN accuracy after one training epoch (Figure 2a), irrespective of their initial performance. Note that this means that after self-supervised fine-tuning, even MP-Net was above all public models, apart from SBERT, in terms of $k$NN accuracy, including models specifically developed to represent scientific texts.

Some of these models (SimCSE, SciNCL, SPECTER, SPECTER2) were originally fine-tuned using the classification token `[CLS]` as sentence representation instead of mean pooling. Therefore, we also measured the $k$NN accuracy using the `[CLS]` representation, before and after contrastive

fine-tuning (Figure 2b); here we used the `[CLS]` representation in the InfoNCE loss function as well. We observed qualitatively the same picture: performance of all models improved with training, sometimes showing even larger improvements (e.g. ∼30 percentage points improvement for SciBERT). However, on average across the 10 models, final performance using mean pooling training was $3.0 \pm 1.0$ percentage points higher compared to using the `[CLS]` token (see Section A.1).

In the literature on self-supervised learning, both in computer vision and in natural language processing, it is common to evaluate representation quality using linear classification accuracy. We evaluated linear classification accuracy for all considered models before and after fine-tuning (Figure 2c) using a logistic regression classifier from `scikit-learn` (Pedregosa et al., 2011). We observed some improvement for most models, apart from the three best-performing citation-informed models. However, we consider this metric less relevant than the $k$NN accuracy for our purposes. Indeed, a representation that has 100% linear classification accuracy but chance-level $k$NN accuracy would be useless for unsupervised tasks such as retrieval, visualisation, or clustering.

### 3.5 CROPPING-BASED FINE-TUNING IS VERY FAST

Using the ICLR dataset, we studied the performance improvement *within* the single fine-tuning epoch. We found that the representation was improved by over 20 percentage points within the first 100 batches (6 400 positive pairs) (Figure 3). Afterwards, the $k$NN accuracy plateaued and did not improve any further, and fine-tuning for more than 1 epoch did not bring further improvements.

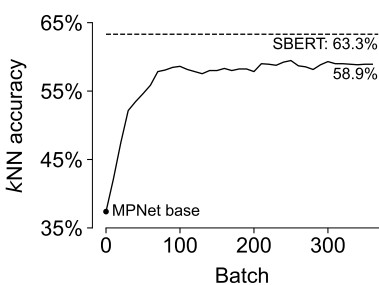

Figure 3: Fine-tuning on ICLR dataset.

These 100 batches of fine-tuning took only ∼1 min of training time on a single GPU (NVIDIA RTX A6000). In comparison, the top-performing sentence transformer models are typically trained on large datasets, with substantial computational costs and training times. For example, the `all-mpnet-base-v2` SBERT model was trained in a supervised way using over one *billion* text pairs. Even though its performance on the ICLR dataset was higher (63.3%), we could bring the same base model (`mpnet-base`) close to SBERT's performance in a few minutes of self-supervised training using five orders of magnitude less data.

## 4 GENERALIZATION PERFORMANCE OF FINE-TUNED MODELS

### 4.1 SENTENCE ADAPTATION IS MORE IMPORTANT THAN DOMAIN ADAPTATION

When fine-tuning a token-level model with a sentence-level contrastive loss on a specific dataset, two distinct mechanisms can contribute to the performance improvement: adaptation to sentence representation, and domain adaptation. To disentangle contributions of these two potential mechanisms, we performed self-supervised fine-tuning on one dataset and assessed the model performance on another dataset from a different domain.

To fine-tune the model, we used biomedical scientific abstracts from the PubMed library (González-Márquez et al., 2024). We used four different sets of PubMed abstracts, all with the same sample size as the ICLR dataset (24 347): surgery abstracts, oncology abstracts, immunology abstracts, and a random selection from the entire PubMed. We fine-tuned the MPNet model for one epoch on each of these four datasets, while evaluating the $k$NN accuracy on the ICLR dataset. After one training epoch, we continued fine-tuning for another epoch on the ICLR dataset itself. We found that the ICLR $k$NN accuracy increased from 37.4% to 56.2–56.8% when training on PubMed data (Figure 4), with almost no difference in performance between the PubMed subsets. Additional training on the ICLR dataset brought the performance up to 58.8–59.8%, which is close to what we reported in Section 3.2 when training on the ICLR dataset directly. We conclude that the majority of the improvement in $k$NN accuracy seen earlier in Table 1 and Figure 2 was due to generic sentence-level adaptation, while domain adaptation had a smaller effect.

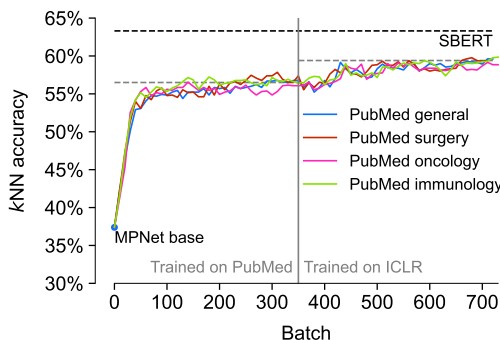

Figure 4: Domain vs. sentence adaptation. $k$NN accuracy on the ICLR dataset during two training epochs: the first epoch used a subset of PubMed abstracts for training, the second epoch used the ICLR dataset. Towards the end of each epoch, the learning saturates and the $k$NN accuracy plateaus (dashed gray lines). SBERT performance is shown for comparison.

Table 2: **Transfer from ICLR fine-tuning to MTEB tasks.** Row blocks correspond to clustering, reranking, retrieval, and STS tasks. All values in percent. Models in columns 4–6 were fine-tuned (or trained) on the ICLR dataset.

| Model | (1) MPNet | (2) SimCSE | (3) SBERT | (4) MPNet | (5) MPNet | (6) Emb. |
|---|---|---|---|---|---|---|
| Pre-trained | yes | yes | yes | yes | yes | no |
| Augmentations | — | — | — | Crops | Dropout | Crops |
| ArxivClusteringP2P | 27.8 | 35.4 | 48.1 | 38.3 | 33.3 | 24.2 |
| BiorxivClusteringP2P | 23.2 | 30.1 | 39.3 | 32.4 | 31.1 | 19.2 |
| MedrxivClusteringP2P | 22.5 | 28.0 | 35.6 | 30.8 | 29.3 | 20.2 |
| RedditClusteringP2P | 37.4 | 44.7 | 56.6 | 55.9 | 49.5 | 20.7 |
| StackExchangeClusteringP2P | 26.3 | 28.8 | 34.3 | 31.3 | 30.2 | 26.3 |
| SciDocsRR | 56.1 | 69.5 | 88.7 | 73.6 | 64.6 | 65.6 |
| MindSmallReranking | 27.5 | 29.3 | 31.0 | 30.2 | 28.4 | 25.9 |
| SCIDOCS | 1.4 | 7.9 | 23.8 | 13.0 | 6.5 | 9.6 |
| ArguAna | 22.2 | 41.4 | 46.5 | 50.6 | 41.9 | 24.1 |
| STS15 | 53.5 | 82.3 | 85.7 | 72.5 | 63.5 | 63.8 |
| STS16 | 50.6 | 77.7 | 80.0 | 76.0 | 66.2 | 56.9 |
| STSBenchmark | 52.0 | 78.6 | 83.4 | 71.7 | 67.9 | 48.8 |
| **Block average** | 33.2 | 46.8 | 55.2 | 48.7 | 42.8 | 35.3 |

## 4.2 GENERALIZATION TO OTHER TASKS

In Section 4.1 we showed that models fine-tuned with cropping augmentations showed good $k$NN performance on other datasets. To assess their performance on other tasks, we evaluated models fine-tuned on the ICLR dataset on several tasks from the Massive Text Embedding Benchmark (MTEB) (Muennighoff et al., 2023). We selected several clustering, retrieval, reranking, and STS tasks (see Section A.2 for details). Clustering tasks assessed the $K$-means clustering results in the embedding space; retrieval and reranking tasks assessed the quality of the $k$NN graph, while the STS tasks measured how well the embedding represents not only small but also large ground-truth pairwise distances.

We found that on average across tasks, SBERT achieved 55.2% performance, MPNet achieved 33.2%, while cropping-based fine-tuning on the ICLR dataset increased the performance to 48.7% (Table 2). This confirmed that cropping-based fine-tuning produced a sentence-level model that showed substantial generalization despite very limited amount of fine-tuning (on ICLR dataset only).

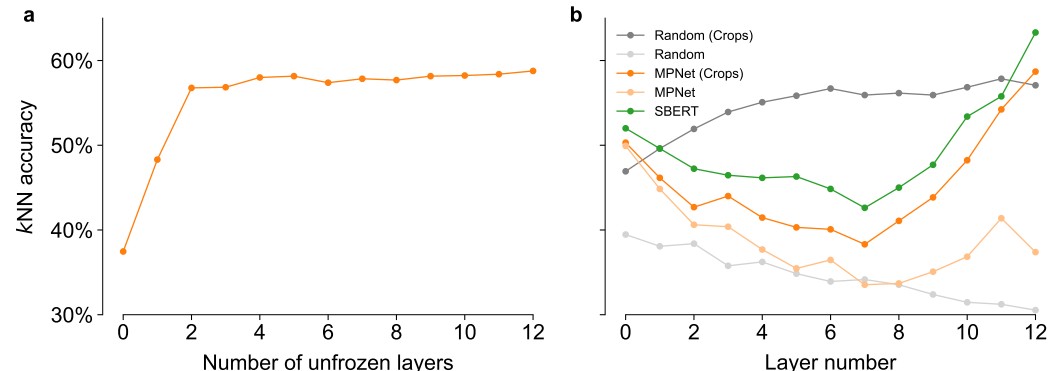

Figure 5: Representation quality across layers. **(a)** $k$NN accuracy after fine-tuning MPNet with different number of initial layers frozen. The embedding layer was frozen in all settings. Zero unfrozen layers corresponds to no fine-tuning. **(b)** $k$NN accuracy after each layer for MPNet before and after fine-tuning, for SBERT, for randomly initialized model, and for model trained with cropping augmentations from scratch. Layer 0 corresponds to the embedding layer.

Note also that some of these datasets, e.g. SCIDOCS, arXiv, and StackExchange, formed part of the *training* set of SBERT, possibly biasing SBERT performance estimates upwards.

As in Section 3.3, cropping-based fine-tuning outperformed dropout-based fine-tuning in all task groups (on average 48.7% vs 42.8%, Table 2, columns 4–5). SimCSE model (which is based on dropout fine-tuning) performed similarly to our dropout-based fine-tuned model on all tasks apart from the STS, suggesting that the good performance of SimCSE in STS benchmarks may be due to some other fine-tuning choices beyond the dropout augmentation or possibly due to substantially larger training dataset (1 M Wikipedia sentences).

Finally, the bare embedding layer trained on the ICLR dataset with cropping-based augmentations (see Section 3.3) showed 35.3% average performance. This unsurprisingly shows that the token embedding model did not generalize well outside of its training domain.

## 5 REPRESENTATION QUALITY ACROSS LAYERS

To investigate whether the entire pre-trained MPNet model needed to be adapted during self-supervised fine-tuning, we performed cropping-based fine-tuning on the ICLR dataset while freezing the embedding layer and various amounts of initial layers. We observed that the performance rapidly improved with the number of unfrozen layers, and fine-tuning only the last two out of 12 layers for one epoch was sufficient to reach almost the same value of $k$NN classification accuracy as fine-tuning the full model (Figure 5a). Unfreezing additional layers led only to minor further improvements.

When all layers were unfrozen, the last few layers underwent the largest change during fine-tuning, while the early layers barely changed, in agreement with previous findings in supervised setting (Merchant et al., 2020; Mosbach et al., 2020). To quantify this, we measured the representation quality after each hidden layer before and after the fine-tuning (Figure 5b). The gap between them was close to zero for early layers and increased towards the last layers. We observed the same effect when fine-tuning MPNet on other datasets (Figure S1).

Intriguingly, the representation quality across layers in our fine-tuned model as well as in SBERT formed a U-shaped curve (Figure 5b): before fine-tuning the embedding layer representation had the highest accuracy, and after fine-tuning it was surpassed by only the last two layers. Across other datasets, the shape was different and not always U-shaped (Figure S1), but fine-tuned models always exhibited a steep rise in performance towards layer 12. The randomly initialized models did not exhibit this shape: after SSL training, the performance monotonically increased and plateaued half-way through the layers (Figure 5b).

Consistently across all datasets and fine-tuned models, the last layer always gave the best representation (Figures 5b and S1). This differs from what has been observed in computer vision, where the top performance after SSL training typically occurs in one of the hidden layers. Indeed, a common practice in computer vision is to have several fully-connected layers (*projection head*) between the output representation and the contrastive loss (Chen et al., 2020), which are discarded after SSL training (*guillotine regularization*) (Bordes et al., 2023). We experimented with adding a one-hidden-layer ($768 \rightarrow 512 \rightarrow 128$) projection head after the average pooling, but this did not consistently affect the quality of the representation, in agreement with Figure 5b.

## 6 DISCUSSION

We showed that self-supervised fine-tuning on a minimal amount of data can lead to large improvements in sentence embedding quality. To this end, we systematically compared different self-supervised augmentation techniques under the exact same training setup and showed that cropping augmentations were much better than dropout augmentations in all evaluations. In fact, cropping augmentations could successfully train a bare embedding layer from scratch to outperform the pre-trained MPNet with dropout fine-tuning. This finding is noteworthy because dropout augmentations of SimCSE (Gao et al., 2021) are arguably the most well-known SSL approach in NLP.

Recent benchmarks found that sentence models that underwent supervised contrastive fine-tuning (based on curated datasets of positive pairs) are superior to self-supervised models (Muennighoff et al., 2023). Here we showed that minimal self-supervised training can improve the quality of sentence embeddings to approach and in some cases almost match the performance of SOTA supervised models, in only a few minutes of training and using five orders of magnitude less data. That said, we did observe a consistent performance gap between our self-supervised fine-tuning and SOTA models like SBERT. Whether this gap is due to the supervised signal or rather to the large amounts of data that SOTA supervised models are trained on, is a topic for future work.

Different metrics have been used in the literature to assess the quality of sentence representations, including linear classification accuracy, nearest-neighbor query performance, and approximating ground-truth semantic textual similarity (STS) (Conneau & Kiela, 2018; Muennighoff et al., 2023). We used the $k$NN accuracy as our main evaluation metric because it is particularly relevant for downstream applications relying on the $k$NN graph, such as retrieval, clustering, and visualization. Many prior works studying sentence representations only evaluated their models on the STS tasks (e.g. Gao et al., 2021; Jung et al., 2024), but previous benchmarks (Muennighoff et al., 2023) and our own results showed that STS performance does not correlate with nearest-neighbor quality. We believe that evaluations similar to $k$NN accuracy should be adopted in benchmarks due to their relevance in practical applications.

While we showed that self-supervised contrastive learning can greatly enhance sentence representations, we believe that there is still a large room for improvement. The lesson from computer vision (Balestriero et al., 2023) as well as our work is that good data augmentations are crucial for the success of self-supervised learning. Combining cropping augmentation with more powerful semantic augmentations such as reformulations using generative language models (Jiang et al., 2022; Wang & Dou, 2023; Abaskohi et al., 2023) can offer an interesting avenue for future research.

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

# A APPENDIX

## A.1 AUGMENTATION AND HYPERPARAMETER CHOICES

In all experiments reported in this section, we repeated the fine-tuning of MPNet for one epoch on the ICLR dataset, and assessed the final $k$NN classification accuracy. In each experiment, all other parameters were kept at their default values described in Section 3.2.

We compared four different pooling strategies for forming the final sentence representation: average pooling, the classification token `[CLS]`, the separation token `[SEP]` (appended at the end of each input text), and the seventh token (as an example of an arbitrary token number). We obtained the best results using the average pooling and `[SEP]` token, with the other two options performing less well (Figure S2a).

Many models evaluated in Section 3.4 also showed higher $k$NN accuracy in the `[SEP]` token representation than using the `[CLS]` token or average pooling (Table S4). Darcet et al. (2023) have recently shown in a computer vision setting that additional tokens can be used by the transformer model as 'registers' to store high-level features. Our results suggest that the same can happen with language models, since the `[SEP]` token often serves as a good sentence representation despite not being explicitly used for training.

**Temperature**   We compared several values of temperature from 0.005 to 5.0, and found that the performance decreased with increasing temperature, with $\tau = 0.005$ and $\tau = 0.05$ yielding similar results (Figure S2a). The value $\tau = 0.5$ used in SimCLR (Chen et al., 2020) performed less well.

**Cropping augmentation**   Our data augmentation consisted of 'cropping out' $t$ consecutive sentences. We varied the number of consecutive sentences (decreasing the batch size accordingly, to make it fit into the GPU memory) and found that the performance generally decreased with $t$, with the optimal number being $t = 2$ (Figure S2b). Note that in our sampling it was possible for the positive pair of text chunks to overlap (but not to coincide exactly).

**Masking augmentation**   We also experimented with a masking augmentation that replaced a certain fraction of tokens in each input chunk with the BERT's special `[MASK]` token. This was done on top of the cropping augmentation. We found that masking led to deterioration of performance (Figure S2c). Using masking augmentation without cropping (i.e. forming positive pairs by applying two different masking patterns to the entire abstract) did not produce competitive results either.

**Learning rate**   The performance increased with increasing the Adam's learning rate (Figure S2d), until it became too large and the training diverged ($\eta \geq 5 \cdot 10^{-4}$). For the bare embedding layer training, the optimal learning rate was $\eta = 5 \cdot 10^{-1}$.

## A.2 MTEB TASKS

**Clustering**   Each of the used datasets consists of texts and ground-truth class labels for each text. The texts are embedded using the model, and the embedding vectors are clustered using a mini-batch $K$-means algorithm with batch size $b = 32$ and $K$ equal to the number of classes. The evaluation score is the so called $V$-measure of agreement between cluster labels and class labels, which is invariant to the permutation of cluster labels.

**Retrieval**   Each of the used datasets consists of a corpus of documents, queries, and a mapping from each query to the relevant documents. The documents and queries are embedded using the model, and the aim is to find the relevant documents within the neighborhood of the query in the embedding space. Neighbors are found using cosine similarity, and after ranking them, normalized discounted cumulative gain ( nDCG) at $k = 10$ nearest neighbors serves as the performance metric.

**Reranking**   Each of the used datasets consists of query texts and a list of relevant and irrelevant reference texts for each query. They are all embedded with the model, and for each query, the text embeddings are ranked based on the cosine similarity to the query embedding. The resulting ranking

is compared to the ground-truth ranking, scored for each query via mean average precision (MAP) metric, and averaged across all queries.

**STS**   Each of the used datasets consists of a set of sentence pairs, each pair with a numerical score from 0 to 5 indicating similarity between the two sentences (5 being most similar, and 0 most dissimilar). All sentences are embedded with the model, and for each pair, the embedding similarity is computed using cosine similarity. These embedding similarities are then compared against ground-truth similarities using Spearman correlation.

For further details, please refer to the original MTEB publication (Muennighoff et al., 2023).

### A.3   SOFTWARE AND DATA

The analysis code is available at `URL`.

## B   SUPPLEMENTARY TABLES AND FIGURES

Table S1: Statistics of the datasets used in the experiments of Table 1. The arXiv, bioRxiv, medRxiv, Reddit, and StackExchange datasets are from the P2P clustering tasks of the Massive Text Embedding Benchmark (MTEB) (Muennighoff et al., 2023), and the ICLR dataset is taken from González-Márquez & Kobak (2024). Length refers to the number of characters. For the arXiv dataset, we used secondary paper categories (e.g., "cs.AI") as labels.

| Dataset | Samples | Classes | Mean length | Std length |
|---|---|---|---|---|
| ICLR | 24 347 | 46 | 1248 | 316 |
| arXiv | 732 723 | 180 | 1010 | 432 |
| bioRxiv | 53 787 | 26 | 1664 | 542 |
| medRxiv | 17 647 | 51 | 1985 | 843 |
| Reddit | 459 399 | 450 | 728 | 710 |
| StackExchange | 75 000 | 610 | 1091 | 809 |

Table S2: $k$NN accuracy using different post-processing transformations of the MPNet mean pooling representation, obtained via the Euclidean and the cosine metrics for finding nearest neighbors, before and after fine-tuning the model.

| | Euclidean | Cosine |
|---|---|---|
| *Before fine-tuning* | | |
| Raw | 37.4% | 39.6% |
| Centered | 37.4% | 37.0% |
| Whitened | 17.6% | 46.9% |
| *After fine-tuning* | | |
| Raw | 58.9% | 59.3% |
| Centered | 58.9% | 58.9% |
| Whitened | 36.2% | 56.5% |

Table S3: Used models.

| Name | Hugging Face | Citation | Year |
|------|-------------|----------|------|
| MPNet | `microsoft/mpnet-base` | Song et al. (2020) | 2020 |
| BERT | `bert-base-uncased` | Devlin et al. (2019) | 2018 |
| SciBERT | `allenai/scibert_scivocab_uncased` | Beltagy et al. (2019) | 2019 |
| SimCSE | `princeton-nlp/unsup-simcse-bert-base-uncased` | Gao et al. (2021) | 2021 |
| DeCLUTR | `johngiorgi/declutr-base` | Giorgi et al. (2021) | 2020 |
| DeCLUTR-sci | `johngiorgi/declutr-sci-base` | Giorgi et al. (2021) | 2022 |
| SciNCL | `malteos/scincl` | Ostendorff et al. (2022) | 2022 |
| SPECTER | `allenai/specter` | Cohan et al. (2020) | 2020 |
| SPECTER2 | `allenai/specter2` | Cohan et al. (2020) | 2022 |
| SBERT | `sentence-transformers/all-mpnet-base-v2` | Reimers & Gurevych (2019) | 2021 |
| Cohere Embed | `embed-english-v3.0` (Cohere API) | `cohere.com` | 2023 |
| OpenAI Embed | `text-embedding-3-large` (OpenAI API) | `openai.com` | 2024 |

Table S4: $k$NN accuracy of different models before our fine-tuning using mean pooling, `[CLS]` token, and `[SEP]` token representations. DeCLUTR(-sci) and SBERT were originally fine-tuned using mean pooling. SimCSE, SciNCL, and SPECTER(2) were originally fine-tuned using the `[CLS]` token. Best representation is in **bold**, best representation for each model is underlined.

| | Average | `[CLS]` | `[SEP]` |
|------|---------|---------|---------|
| MPNet | 37.4% | 31.8% | 36.3% |
| BERT | 40.6% | 28.2% | 33.1% |
| SciBERT | 44.5% | 21.5% | 28.5% |
| SimCSE | 45.7% | 43.5% | 46.4% |
| DeCLUTR | 50.3% | 45.0% | 34.8% |
| DeCLUTR-sci | 53.5% | 38.8% | 29.2% |
| SciNCL | 57.0% | 56.8% | 57.8% |
| SPECTER | 56.8% | 54.1% | 58.5% |
| SPECTER2 | 58.2% | 57.2% | 59.7% |
| SBERT | **63.3%** | 56.8% | 59.8% |

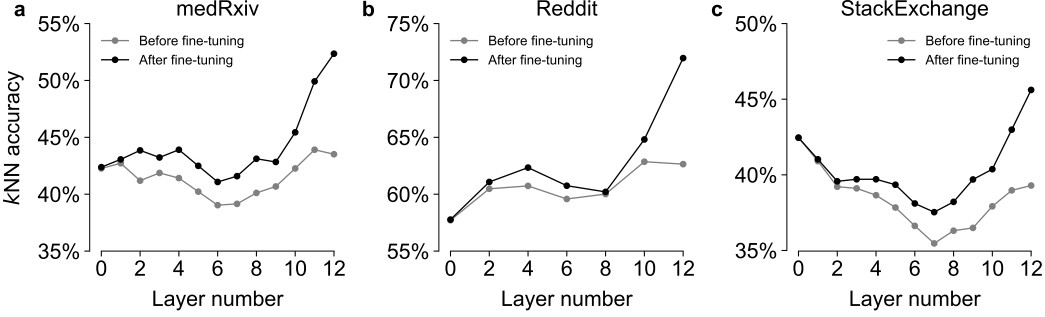

Figure S1: Representation quality across layers. $k$NN accuracy after each layer for MPNet before and after fine-tuning in the (a) medRxiv, (b) Reddit, and (c) StackExchange datasets. Evaluation is also done on the respective dataset.

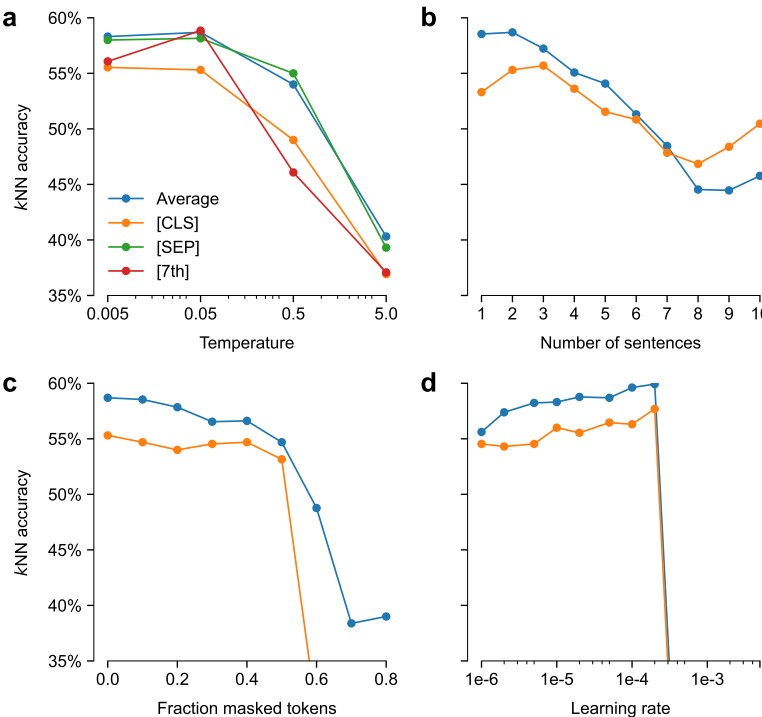

Figure S2: Hyperparameter tuning. **(a)** Temperature $\tau$ used to scale the similarities in the loss function. **(b)** Number of consecutive sentences $t$ used in the cropping augmentation. The minibatch size $b$ was adapted depending on $t$ to make it fit into our GPU memory: we used $b = 128$ for $t = 1$; $b = 64$ for $t = 2, 3, 4$; $b = 32$ for $t = 5, 6, 7, 8, 9$; and $b = 16$ for $t = 10$. **(c)** Fraction of masked tokens used in addition of the cropping augmentation. **(d)** Learning rate $\eta$ used by the Adam optimizer.

