# OpenReview forum: "Is self-supervision enough for training sentence embeddings?"
_ICLR.cc/2025/Conference — Submitted to ICLR 2025_

### Official Review · Reviewer_6XYP · 2024-11-02

**Soundness:** 2
**Presentation:** 2
**Contribution:** 2
**Rating:** 5
**Confidence:** 3

**Summary:**

This paper investigates whether self-supervised learning (i.e., using the domain corpus only) alone can produce high-quality sentence embeddings without extensive supervised fine-tuning (i.e., using a sentence pair dataset).
The authors re-explore the effectiveness of cropping-based augmentation for contrastive learning, demonstrating that this approach performs better than traditional dropout-based augmentation, SimCSE.
The paper shows that sentence embeddings can achieve comparable performance with supervised fine-tuning in some embedding tasks.
Key contributions include a systematic comparison of augmentation strategies and evidence that self-supervision can suffice for strong sentence representations.
This work challenges the need for supervised data in sentence embedding and offers insights for more efficient embedding training.

**Strengths:**

The paper demonstrates that self-supervised learning alone can produce high-quality sentence embeddings, reducing dependence on supervised data.

It systematically compares augmentation techniques, highlighting the effectiveness of cropping-based augmentation over traditional dropout methods used in SimCSE.

**Weaknesses:**

The research question "Is self-supervision enough for training sentence embeddings?" is not convincingly answered. While there are various experimental results in the paper, the insights derived from them seem somewhat lacking in coherence and strength with respect to the research question.

While the paper provides experimental findings, it lacks deep discussions or analyses about why it happens. For example, although the experimental results about the superior performance of cropping are somehow different from the conclusion in the SimCSE paper [Gao et al. 2021], there is no discussion or investigation about the reason. Gao et al. [2021] show that SimCSE is better than cropping and the next sentence pairing [Logeswaran and Lee, 2018].

**Questions:**

>Note that we purposefully used the entire dataset first for self-supervised training and later for supervised evaluation. As SSL does not have access to class labels, this does not present overfitting issues.  For supervised evaluation, we used a 9:1 train/test split (using only labeled data).

Did the self-supervised training use test-split data too? If yes, the setting is a little tricky, and "As SSL does not have access to class labels, this does not present overfitting issues" is misleading. Even without access to the class labels, it should be a dataset leakage or a transductive learning setting.


>our own results showed that STS performance does not correlate with nearest-neighbor quality

Where is the result?


Does this self-supervised contrastive approach have any limitations? For example, is there any task where self-supervised contrastive learning may underperform supervised learning in MTEB or other tasks using embeddings?


Pre-training (e.g., language modeling) is also self-supervised learning. The title is misleading because it is ambiguous. Please consider different titles or wording to clarify it.

---

> ### Author Response · Authors · 2024-11-22
>
> We would like to thank the reviewer for their helpful feedback.
>
> > [...] although the experimental results about the superior performance of cropping are somehow different from the conclusion in the SimCSE paper [Gao et al. 2021], there is no discussion or investigation about the reason. Gao et al. [2021] show that SimCSE is better than cropping and the next sentence pairing [Logeswaran and Lee, 2018].
>
> In the SimCSE paper the authors indeed claim that dropout augmentations are better than crop augmentations; however, they only evaluate on STS tasks. Later benchmarks (e.g. MTEB benchmark paper) have found that performance on STS tasks does not correlate well with performance on other tasks, and that SimCSE model does not perform very well across the tasks. The main point in our work was to compare these two augmentation strategies under the exact same training setting, and we observed that dropout augmentations (as well as the out-of-the-box SimCSE model) perform substantially worse on all tasks except for the STS (in fact, our dropout fine-tuning did not lead to a good performance on STS either; it is unclear why the out-of-the-box SimCSE model performs better on STS). Apart from that, the SimCSE paper does not report any details on their "crop augmentation" setup, making it difficult to investigate this difference further.
>
> Given the popularity and the fame of SimCSE paper, we consider our dropout-vs-crops comparison an important contribution.
>
> > Did the self-supervised training use test-split data too? If yes, the setting is a little tricky
>
> Yes it did. The same concern was raised by another reviewer as well, so to demonstrate that this is not a problem, we repeated our experiments, this time excluding the test set from the SSL training data. The results almost did not change at all. In this table, the "Dropout" and "Crops" columns show results for SSL training on the full data (our original results), and the columns with "(train)" show results for SSL training only on the train set.
>
> |                 |   Dropout |   Dropout (train) |   Crops |   Crops (train) |
> |:----------------|----------:|------------------:|--------:|----------------:|
> | ICLR            |      46.8 |              46.7 |    58.9 |            56.8 |
> | arxiv           |      39.9 |              38.9 |    44.2 |            44.1 |
> | biorxiv         |      60.7 |              60.9 |    61.8 |            61.6 |
> | medrxiv         |      47.8 |              47.8 |    52.4 |            52.3 |
> | reddit          |      57.8 |              59.6 |    72.0 |            71.8 |
> | stackexchange   |      41.6 |              41.3 |    45.6 |            45.1 |
>
> Given that this was raised as a repeated concern, we would be happy to switch to this kind of evaluation.
>
> > "We show that STS performance does not correlate with nearest-neighbor quality". Where is the result?
>
> We evaluated kNN accuracy of the out-of-the-box SimCSE model and the results were poor on all tasks apart from the STS tasks (see Tables 1 and 2). We will reformulate this sentence to clarify.
>
> Apart from that, we will add a Limitations section (as an example, our SSL training does not surpass SBERT, and it remains for future work to understand why exactly). We will also consider alternative titles, however we think that having "sentence embeddings" in the title already makes it unambiguous.

---

### Official Review · Reviewer_XULo · 2024-11-04

**Soundness:** 2
**Presentation:** 4
**Contribution:** 2
**Rating:** 5
**Confidence:** 4

**Summary:**

This paper proposes fine-tuning sentence embeddings on consecutive text crops sampled from a dataset to improve embedding performance on that same dataset. The fine-tuning objective is InfoNCE with cosine as the similarity metric.  The evaluation is done on MTEB clustering tasks, recast as knn accuracy. The results indicate that, for a particular type of model (MPNet), text crops work much better than alternative strategies, such as dropout augmentation. The approach does not perform as well as SBERT. Further analysis leads to claims of generalization and looks at the performance at different layers.

**Strengths:**

1. Well written and easy to follow.
2. Thorough analysis and sound experiments.
3. Sufficient substance, including detailed appendix.

**Weaknesses:**

1. Overclaiming in the central claim: The title is honestly just completely wrong — the work is not about whether SSL is “enough” for training sentence embeddings. The work, if I put it bluntly, proposes a tried and trusted method in machine learning: tuning on the test set. Please correct me if I’m wrong, but the approach fine-tunes on the same dataset that it is evaluated on, using essentially the same objective, so it is no surprise that the method outperforms its (weak) baseline. I suspect that alternative approaches (eg finetuning pretrained BERT NSP/MLM-style on the same dataset) would also see gains. The authors argue that given that “SSL does not have access to class labels”, there are no “overfitting issues” — this is misguided: the central claim is directly related to “having access” to the test set. It would be really worrying if this method did not see improved performance. Why not do K-fold cross-validation to show that the approach holds up generally, with confidence bounds that show that differences in performance are statistically significant?

2. Lack of performance: Despite tuning on the test set, the approach does not outperform SBERT. What is the value of applying this method if I could simply apply SBERT? If the approach really works, couldn’t we apply it to SBERT to get SOTA? That would be valuable to practitioners. Similarly, you could evaluate whether downstream performance (eg in a RAG pipeline) improves if you adapt the embeddings to the domain.

3. Overclaiming in adjacent claims: The first sentence of the discussion “We showed that self-supervised fine-tuning on a minimal amount of data can lead to large improvements in sentence embedding quality” should be rewritten to “We showed that self-supervised fine-tuning on a given dataset improves sentence embedding quality on that dataset”. It is a much narrower claim. It is important to be precise. In l.375-77 “We conclude that the majority of the improvement was due to generic sentence-level adaptation, while domain adaptation had a smaller effect” — it’s still a similar domain, scientific literature clustering? This claim is much too strong. Related to the first point above, if you want to make this claim, you need to provide much more and much stricter evidence.


Minor:
- The correct citations for sentence representations/embeddings would be “SkipThought” (Kiros et al; 2015) and “Distributed Representations of Sentences” (Hill et al; 2016), definitely not Reimers et al.

**Questions:**

1. Why train using cosine but evaluate using Euclidean?
2. How was the learning rate chosen? Wouldn’t it be better to tune with a range of learning rates and pick the average (or best if you had a validation set)? How do I know that your dropout results are not poor simply because you selected your learning rate based on what worked best for cropping?
3. “It is unclear how their [SSL approaches] compares between each other and to the SOTA” (l. 43) — what is unclear and why? It’s phrased in a way where it needs a better explanation, and it’s not actually something you are really examining in this paper?
4. Why was MPNet chosen as the model to do the experiments with? No justification is given. I’d think it would make the paper stronger if you can show the same results for different models.
5. Specifically, how does this method perform when you apply it to SBERT rather than MPNet? If it improves SBERT, then this could be a valuable contribution?
6. What happens in section 4.1 when you use something like CCNews (or something even more out-of-domain) rather than PubMed?

---

> ### Author Response · Authors · 2024-11-22
>
> We would like to thank the reviewer for very helpful feedback.
>
> > The title is honestly just completely wrong — the work is not about whether SSL is “enough” for training sentence embeddings. The work, if I put it bluntly, proposes a tried and trusted method in machine learning: tuning on the test set. Please correct me if I’m wrong, but the approach fine-tunes on the same dataset that it is evaluated on, using essentially the same objective, so it is no surprise that the method outperforms its (weak) baseline.
>
> There may be some confusion here. The SSL objective that we used for training and the evaluation metric are very different: the objective  brings sentences from the *same* text close together in the representation space, without any use of class labels, while the evaluation metric quantifies how close *different* texts of the same class are.
>
> To demonstrate that there was no overfitting, we repeated our SSL training process using only the training set, subsequently trained the kNN classifier again only on the training set, and finally evaluated the classifier on the test set. This way, the test set was never used for the SSL training. As you can see in the table below, this made almost no difference. In this table, the "Dropout" and "Crops" columns show results for SSL training on the full data (our original results), and the columns with "(train)" show results for SSL training only on the train set.
>
> |                 |   Dropout |   Dropout (train) |   Crops |   Crops (train) |
> |:----------------|----------:|------------------:|--------:|----------------:|
> | ICLR            |      46.8 |              46.7 |    58.9 |            56.8 |
> | arxiv           |      39.9 |              38.9 |    44.2 |            44.1 |
> | biorxiv         |      60.7 |              60.9 |    61.8 |            61.6 |
> | medrxiv         |      47.8 |              47.8 |    52.4 |            52.3 |
> | reddit          |      57.8 |              59.6 |    72.0 |            71.8 |
> | stackexchange   |      41.6 |              41.3 |    45.6 |            45.1 |
>
> Does this address your concern? The same point was raised by another review, and so to avoid the confusion, we would be happy to switch to this kind of training/evaluation setup.
>
> > [...] the approach does not outperform SBERT. What is the value of applying this method if I could simply apply SBERT?
>
> The goal of our work was not to propose a new approach or model outperforming others, but to study self-supervised training process for sentence embeddings and to systematically compare the two most popular augmentation approaches (which to the extent of our knowledge have not been systematically compared to date and have been reported in the literature as yielding conflicting results).
>
> > If the approach really works, couldn’t we apply it to SBERT to get SOTA?
>
> This we have already done in the paper, at least for the ICLR dataset: shown by the green arrow in Figure 2 (change in kNN accuracy when finetuning SBERT with crop-based SSL). We have now repeated this experiment for all other MTEB datasets from Table 1 and the results are similar: SBERT representation quality does not improve after fine-tuning with crop augmentations.
>
> |               |   SBERT (out-of-the-box)|   SBERT (after fine-tuning) |
> |:--------------|--------:|----------------:|
> | ICLR          |    63.3 |            61.5 |
> | arxiv         |    46.2 |            45.2 |
> | biorxiv       |    65.2 |            64.9 |
> | medrxiv       |    56.8 |            54.1 |
> | reddit        |    75.0 |            75.3 |
> | stackexchange |    50.6 |            51.0 |
>
> But again, our goal was not to improve on SBERT, but rather to study how close and how fast can one get to its performance using *only* self-supervised learning (and what changes occur in the model during this fine-tuning).
>
>
> > In l.375-77 “We conclude that the majority of the improvement was due to generic sentence-level adaptation, while domain adaptation had a smaller effect” — it’s still a similar domain, scientific literature clustering? This claim is much too strong.
>
> This is a fair point, and we agree that the PubMed dataset is not entirely "out of domain". We have now repeated this experiment using the Reddit dataset which is arguably more "out of domain". As the reviewer expected, this yielded slightly worse performance on ICLR compared to the training on PubMed dataset: with PubMed, we obtained 56.8% accuracy, while with Reddit we obtained 53.5%. However, this is still a large improvement compared to the starting MPNet performance (37.4%), so we still believe that the majority of the improvement is due to sentence adaptation (~75% of the improvement in this new experiment). We will include this experiment into the paper and rephrase this section accordingly.

---

> ### Author Response · Authors · 2024-11-22
>
> **Minor comments**
>
> > Why train using cosine but evaluate using Euclidean?
>
> The Euclidean and cosine evluations gave very similar results (see Table S2, l. 843). We would be happy to switch to cosine evaluation in future revisions.
>
> > How was the learning rate chosen? [...] How do I know that your dropout results are not poor simply because you selected your learning rate based on what worked best for cropping?
>
> See supplementary Section A.1 on hyperparameter choices. We have now assessed the performance of dropout augmentation with various learning rates, and found that it varied very little and was always substantially worse than with crops. We will add it to the paper.
>
> > Why was MPNet chosen as the model to do the experiments with? No justification is given. [...] I’d think it would make the paper stronger if you can show the same results for different models
>
> As written on line 118, we chose MPNet because it is the base model used by SBERT. Please note that MPNet is not fundamentally different from any other BERT-based model. We applied our fine-tuning to many other models as shown in Figure 2 and described in Section 3.4.
>
> Apart from that, we are happy to include suggested citations and adjust the introduction and discussion formulations, as suggested.

---

> > ### Comment · Reviewer_XULo · 2024-11-25
> >
> > Thank you for the detailed response. I have revised my score. That said, if I read the other reviews I think we share similar concerns around the preciseness and substantiation of your claims.

---

> > > ### Author Response · Authors · 2024-11-25
> > >
> > > Thank you. We would very much appreciate if you could comment a bit more on the training/test split issue, as it could help us in future revisions. Does the train/test separation (first bullet point in our reply above) fully address your concern (first bullet point in your review)? If we use this kind of training & evaluation, then is this issue entirely resolved for you? Or was your concern different? Thanks.

---

> > > > ### Comment · Reviewer_XULo · 2024-11-25
> > > >
> > > > Yes, if you train on the train set and test on the test set, that concern is addressed.

---

> > > > > ### Author Response · Authors · 2024-11-27
> > > > >
> > > > > Thanks, that's good to hear. If you could elaborate on what are the critical issues that remain unaddressed for you, it would be helpful for us. Your point (1) is addressed, your point (3) we hope is also addressed by our additional experiments; see above about point (2).
> > > > >
> > > > > PS. It would be great to hear from the other two reviewers as well. Thanks!

---

### Official Review · Reviewer_aLEN · 2024-11-08

**Soundness:** 2
**Presentation:** 2
**Contribution:** 2
**Rating:** 3
**Confidence:** 4

**Summary:**

The paper investigates the potential of self-supervised learning (SSL) for generating high-quality sentence embeddings without relying on large, supervised datasets. The authors evaluate various SSL techniques, particularly focusing on text crops as a data augmentation strategy, and find it outperforms traditional dropout-based methods. Their findings suggest that SSL alone can yield competitive embeddings, with performance close to supervised models, and highlight that most improvements stem from generic sentence adaptation rather than domain adaptation. They also emphasize that embeddings can perform well even when trained from scratch or with minimal architecture.

**Strengths:**

- The paper compares multiple self-supervised learning approaches for sentence embeddings, providing insights into the effectiveness of different data augmentation strategies, especially text crops.

- It demonstrates that SSL can achieve sentence embedding quality close to supervised models, suggesting potential for relying more on SSL instead of supervised fine-tuning on large datasets.

**Weaknesses:**

- The paper lacks originality, as most of its observations are already well-known in the research community. Self-supervised learning (SSL) has long been applied for training embeddings, and data augmentation techniques like text cropping and dropout are established as beneficial. The paper does not introduce new techniques or offer novel insights in this area.

- The study does not compare with current state-of-the-art models, such as BAAI's BGE models or commercial models (such as OpenAI and Voyage AI). Although it’s true that these models may be trained on different data, this lack of such results weakens the paper’s contribution. Its conclusions would be far more convincing if the proposed model was trained on the same or similar data as state-of-the-art models.

- Additionally, the evaluation is limited, omitting several popular benchmarks; for instance, many datasets from the MTEB benchmark are not included in the analysis.

**Questions:**

N/A

---

> ### Author Response · Authors · 2024-11-22
>
> We thank the reviewer for their comments.
>
> > The paper lacks originality, as most of its observations are already well-known in the research community [...]
>
> The point of the paper is not to introduce a new technique, but to systematically compare the existing augmentation techniques under the exact same training setup (model, data, hyperparams, etc.). Among other things, we find that the cropping augmentation outperforms dropout augmentation (SimCSE) by a very large margin. We believe this is not a well-known result. If the reviewer believes otherwise, we would very much appreciate a reference.
>
> In fact, another reviewer asked why our results contradict the claims given in the SimCSE paper. *Our results cannot possibly be both "well-known" and contradictory or surprising given the literature!* In fact, given the popularity and the fame of the SimCSE paper, we consider our dropout-vs-crops comparison an important contribution.
>
> > The study does not compare with current state-of-the-art models, such as BAAI's BGE models or commercial models (such as OpenAI [...]
>
> Actually we did compare with state-of-the-art models, including commercial models such as `embed-english-v3.0` by Cohere and `text-embedding-3-large` by OpenAI (see line 309). We found that on the ICLR dataset they performed similar to SBERT. Following the reviewer's suggestion, we now also evaluated BAAI's FlagEmbedding model (`BAAI/llm-embedder`) on the ICLR dataset and it yielded an accuracy of 62.7%, making it not much different from other commercial models and SBERT.
>
> > Its conclusions would be far more convincing if the proposed model was trained on the same or similar data as state-of-the-art models.
>
> Most commercial models do not disclose which data they used for training. Furthermore, most SOTA models (including SBERT) are trained with supervised positive pairs, whereas our paper is explicitly about self-supervised learning only. Doing a comparison of the self-supervised and supervised contrastive training would be interesting, but this is out of the scope of our work; we discussed this idea in our discussion (line 510) and it is an interesting future research direction.
>
> > the evaluation is limited, omitting several popular benchmarks; for instance, many datasets from the MTEB benchmark are not included
>
> We did not include all MTEB tasks in our paper for simplicity, but instead, chose multiple tasks from each evaluation modality of MTEB (clustering, retrieval, etc.). In addition to that, we evaluated kNN classification accuracy which is not even included into MTEB. We would be happy to add any specific evaluation, if the reviewer has concrete suggestions.

---

### Meta-Review · Area_Chair_VReW · 2024-12-23

**Metareview:**

The paper explores the potential of self-supervised learning to train sentence embeddings without supervised fine-tuning, systematically comparing augmentation strategies like text cropping and dropout. A key strength is its thorough experimental analysis, demonstrating that cropping significantly outperforms dropout in contrastive learning and that SSL can achieve embedding quality close to supervised methods. However, the paper is criticized for its lack of originality, as it mainly revisits existing techniques, and for overclaiming its findings, particularly regarding generalizability and domain adaptation. Additionally, it does not outperform state-of-the-art supervised models like SBERT, and its evaluation is limited, omitting key datasets. Two reviewers expressed concerns about using test sets in the SSL setup, though the authors provided clarifications and additional experiments to address this. The reviewers generally lean towards rejection of this paper with scores 3,5,5. I agree with the reviewers and think overclaiming is a serious issue, and this paper needs significant revisions to address that to be published.

**Additional Comments On Reviewer Discussion:**

The reviewers acknowledged the paper’s systematic comparison of augmentation techniques and its demonstration that SSL can produce strong sentence embeddings, but they criticized its lack of originality, overclaiming of results, limited evaluation benchmarks, and use of test data in SSL training. Reviewers questioned why the approach did not outperform SBERT and asked for broader model and task evaluations. The authors clarified that the goal was not to surpass SBERT but to study SSL performance, conducted additional experiments excluding test data from SSL training, and added evaluations on out-of-domain datasets. I think overclaiming is a serious issue and I also agree that more evaluations need to be added, for example, even though the authors claim that the goal of this paper was not to surpass some supervised models, the paper title does not seem to be in this case.

---

### Decision · Program_Chairs · 2025-01-22

Reject